# Mitochondrial Disease (MELAS Syndrome) Discovered at the Start of Pregnancy in a Patient with Advanced CKD: A Clinical and Ethical Challenge

**DOI:** 10.3390/jcm8030303

**Published:** 2019-03-04

**Authors:** Domenico Santoro, Gianluca Di Bella, Antonio Toscano, Olimpia Musumeci, Michele Buemi, Giorgina Barbara Piccoli

**Affiliations:** 1Nephrology and Dialysis Unit, Department of Clinical and Experimental Medicine, University of Messina, 98100 Messina, Italy; dsantoro@unime.it (D.S.); buemim@unime.it (M.B.); 2Department of Clinical and Experimental Medicine and Pharmacology, University of Messina, 98100 Messina, Italy; gianluca.dibella@tiscali.it; 3Neurology, Department of Clinical and Experimental Medicine, University of Messina, 98100 Messina, Italy; toscano.antonio@unime.it (A.T.); olimpia.musumeci@unime.it (O.M.); 4Department of Clinical and Biological Sciences, University of Torino, 10100 Torino, Italy; 5Néphrologie, Centre Hospitalier Le Mans, 72000 Le Mans, France

**Keywords:** chronic kidney disease, pregnancy, focal segmental glomerulosclerosis, MELAS syndrome, mitochondrial diseases

## Abstract

Pregnancy is a challenge in the life of a woman with chronic kidney disease (CKD), but also represents an occasion for physicians to make or reconsider diagnosis of kidney disease. Counselling is particularly challenging in cases in which a genetic disease with a heterogeneous and unpredictable phenotype is discovered in pregnancy. The case reported regards a young woman with Stage-4 CKD, in which “Mitochondrial Encephalopathy, Lactic Acidosis, and Stroke-like episodes” (MELAS syndrome), was diagnosed during an unplanned pregnancy. A 31-year-old Caucasian woman, being followed for Stage-4 CKD, sought her nephrologist’s advice at the start of an unplanned pregnancy. Her most recent data included serum creatinine 2–2.2 mg/dL, Blood urea nitrogen (BUN) 50 mg/dL, creatinine clearance 20–25 mL/min, proteinuria at about 2 g/day, and mild hypertension which was well controlled by angiotensin-converting enzyme inhibitors (ACEi); her body mass index (BMI) was 21 kg/m^2^ (height 152 cm, weight 47.5 kg). Her medical history was characterized by non-insulin-dependent diabetes mellitus (at the age of 25), Hashimoto’s thyroiditis, and focal segmental glomerulosclerosis. The patient’s mother was diabetic and had mild CKD. Mild hearing impairment and cardiac hypertrophy were also detected, thus leading to suspect a mitochondrial disease (i.e., MELAS syndrome), subsequently confirmed by genetic analysis. The presence of advanced CKD, hypertension, and proteinuria is associated with a high, but difficult to quantify, risk of preterm delivery and progression of kidney damage in the mother; MELAS syndrome is per se associated with an increased risk of preeclampsia. Preterm delivery, associated with neurological impairment and low nephron number can worsen the prognosis of MELAS in an unpredictable way. This case underlines the importance of pregnancy as an occasion to detect CKD and reconsider diagnosis. It also suggests that mitochondrial disorders should be considered in the differential diagnosis of kidney impairment in patients who display an array of other signs and symptoms, mainly type-2 diabetes, kidney disease, and vascular problems, and highlights the difficulties encountered in counselling and the need for further studies on CKD in pregnancy.

## 1. Background

Pregnancy represents a critical moment in the life of a woman with chronic kidney disease (CKD). On the one hand, a successful pregnancy enables a woman to demonstrate that living with CKD is compatible with attaining an important milestone in life [1]. On the other, pregnancy is an occasion for reconsidering diagnosis and treatment of CKD, and, as in the case described, the definition of a genetic disease may pose difficult clinical and ethical problems.

Before and during pregnancy, counselling of patients with CKD, as well as with other chronic diseases, needs to take into account the effect the disease is likely to have on pregnancy outcomes and on the foetus and the effect of pregnancy on the prognosis of the disease in the mother [2,3,4].

In the case of CKD, available data suggest that pregnancy has no detrimental effect on progression if CKD is in the early stages, but that it may hasten progression of CKD when it occurs in advanced stages [4,5,6,7]. Furthermore, proteinuria and hypertension can have negative effects on pregnancy outcomes and may be associated with disease progression in the mother [5,6,7].

Overall, the main risks for the offspring are those linked to prematurity, and may affect neurological development, as well as long term health [8,9,10]. The situation is even more complex in the case of hereditary kidney diseases, in particularly if they develop in adulthood and present phenotypic heterogeneity [11]. The case reported is an example, as it regards a young woman with stage-4 CKD, in which the diagnosis of a complex mitochondrial hereditary disease, MELAS syndrome, an acronym for “Mitochondrial Encephalopathy, Lactic Acidosis, and Stroke-like episodes” was made during an unplanned pregnancy.

## 2. The Case

A 31-year-old Caucasian woman, who worked as a nurse in a retirement home and was on regular follow-up for Stage-4 CKD, sought her nephrologist’s advice at the start of an unplanned pregnancy, worried about the risk of rapid worsening of her residual kidney function, and the ways her kidney disease might affect the child. 

Her most recent kidney function data included serum creatinine 2–2.2 mg/dL, blood urea nitrogen (BUN) 50 mg/dL, creatinine clearance 20–25 mL/min, between 1 and 2 g/day, mild hypertension which was well controlled by angiotension-converting enzyme inhibitors (ACEi), which were discontinued at the time of the positive pregnancy test (self-performed commercial rapid test, at 6 weeks of gestation, later confirmed by echography at 8 weeks of gestation). 

The woman’s medical history was characterized by early onset of non-insulin-dependent diabetes mellitus (at the age of 25), followed by Hashimoto’s thyroiditis at the age of 26. At the age of 30, one year prior to becoming pregnant, during a workup for infertility (ascribed to bilateral fallopian tube obstruction), she was found to have increased creatinine levels and proteinuria. A kidney biopsy was performed, and disclosed focal segmental glomerulosclerosis, with advanced sclerotic lesions (Figure 1). The patient’s mother was affected by Type-2 diabetes and mild CKD but the family history was otherwise uneventful.

Treatment with ACEi and nutritional management was started, in the hope of delaying the progression of kidney disease and controlling proteinuria. At the time of the unexpected conception, the woman was in the initial stages of being evaluated for pre-emptive kidney transplantation. 

Physical examination revealed a young, apparently healthy woman, of small body size and normal body mass index (BMI) (height 152 cm, weight 47.5 kg, BMI: 21 kg/m^2^). There were no other pathologic findings, except for a mild hearing impairment. 

Electrocardiography suggested cardiac hypertrophy, confirmed at cardiac ultrasound, organised in the context of evaluations for kidney transplantation, which disclosed signs of relevant cardiac hypertrophy, out of proportion with her mild hypertension, confirmed at cardiac magnetic resonance (Figure 2, Figure 3 and Figure 4).

The association of short stature, kidney disease, diabetes, cardiac hypertrophy and hearing loss strongly supported a diagnosis of genetic disease, in particular the possibility of a mitochondrial cytopathy (MELAS syndrome). This was confirmed by genetic analysis (Sanger sequencing of *MT-TL1* gene from DNA extracted from blood and urine), which disclosed a mutation in the mitochondrial gene *MT-TL1* (MIM ID *590050-0001), encoding mitochondrial tRNA leucine 1 (3243A > G transition). Specific questioning brought to light a history of nausea and gastroenteric disturbances, also frequently associated with MELAS.

The risks associated with pregnancy in the late CKD stages, the clinical implications of the diagnosis of MELAS, and the difficulty in foreseeing negative consequences for the offspring were extensively discussed and the patient and her husband decided to terminate pregnancy, given the high probability of hastening the progression of CKD, the unpredictable prognosis for the child, and, even more importantly, for the mother. After pregnancy termination, therapy with ACEi was started again. Kidney function remained stable up to two years after pregnancy discontinuation (at February 2019: serum creatinine 2.0 mg/dL, proteinuria 2.16 g/24 h).

## 3. Discussion: Diagnostic Issues

The diagnosis of a mitochondrial disease remains a clinical challenge. In spite of increasing interest in mitochondrial dysfunction in kidney diseases, in clinical practice the diagnosis of mitochondrial cytopathies commonly eludes detection [12,13,14]. 

The acronym MELAS summarizes the syndrome’s most common neurologic presentation (Mitochondrial Encephalopathy, Lactic Acidosis, and Stroke-like episodes) [15,16]. However, the presenting picture may be neurologically silent or may include an array of signs and symptoms appearing either together or at distance one from the other. These include various kidney diseases, cardiac hypertrophy, short stature, below average intellectual ability, and juvenile-onset type-2 diabetes [14,15]. Hearing loss is frequent. The onset of each of these problems is usually gradual, and this may further hasten diagnosis; however, a sudden onset of one of these deficits is not exceptional, in keeping with the microvascular ischemic pathogenesis. 

Over time, the disease often progresses to ischemic stroke and dementia, sometimes with outbursts of anger. It is now thought, for example, that Nietzsche’s mental instability was caused by MELAS syndrome (16). The most common mutation, present in about 80% of cases, was the one present in our patient. Other mutations are increasingly being described in the literature [14,15].

The kidney disease associated with MELAS syndrome is protean and may involve at a different degree all structures: vascular (infarction, nephroangiosclerosis), interstitial (Fancony syndrome, complex tubular disorders), and glomerular, where the appearance of focal segmental lesions is frequently associated with progressive kidney failure [17,18,19,20]. Different kidney neoplastic diseases may also occur [18].

The kidney biopsy in this case presents the features of glomerular sclerosis and of glomerulomegalia, thus suggesting a form of focal segmental glomerulosclerosis (FSGS), secondary to nephron loss and to hypertrophy of the remnant nephrons (Figure 1).

The renal involvement in mitochondrial disorders is relatively common and may take different forms; heterogeneity is high both in the context of the same mutation (the best known of which is the one presented by our patient, characteristic of the MELAS syndrome), where probably the main determinant is the quantity of mutated DNA in each tissue, and across the different mutations described.

Two other well-characterised diseases with kidney involvement are Kearns-Sayre syndrome (KSS) characterized by early onset (before 20 years of age) of progressive external ophthalmoplegia and pigmentary retinitis; deafness, cerebellar ataxia, and heart block are frequent features, and kidney disease is frequent. In these cases, the involvement is typically interstitial, with renal tubular acidosis, Barter syndrome, or tubulo-interstitial nephritis, with or without nephrocalcinosis, while focal segmental glomerulosclerosis is rare. The so-called “Mitochondrial depletion syndromes” are characterised by a reduction in the quantity of mtDNA within a mitochondrion of a cell; these heterogeneous diseases are linked to various mutations, and are likewise associated with interstitial kidney diseases. Kidney failure however, seems rare.

While an extensive review is beyond the scope of this paper, the readers may refer to the brilliant review by Finsterer and Scorza for an extensive discussion on the kidney manifestations of mitochondrial diseases [21].

The disease is transmitted by the mother, does not skip generations, and affects both sexes. Typically, symptoms vary in members of the family. Phenotype differences are considered as mainly linked to the different charge of mutations in the mitochondria, in the different tissues, but mutations are ubiquitous. Pre-implantation selection is now being occasionally used with positive results [22,23].

In our case, in keeping with phenotypic heterogeneity, there was a family history of diabetes and the patient’s mother had mild CKD. No severe neurologic deficit was evident at presentation in either the patient or her mother.

There is no treatment for this protean disease, although L-arginine, carnitine and coenzyme Q10 have been employed with promising results [24].

## 4. Discussion: Counselling Issues

It is hard to imagine a more difficult counselling task than telling a young woman who is already stunned by the start of an unplanned pregnancy that the re-evaluation of her nephrology history demonstrates the presence of a genetic, probably progressive disease, that could lead not only to end stage kidney disease, but also to dementia, that the same disease would be inherited by her child, and, at the same time, that it is presently impossible to foresee the child’s phenotype and progression of the disease in herself. 

The classic indications for counselling in CKD pregnancy are to consider the health of the mother and of the foetus separately, explaining the uncertainties for each party, and the potential interactions between maternal issues and the child’s short- and long-term prognosis [3,4].

Independently from the diagnosis for the mother, the combination of reduction in kidney function, hypertension, and proteinuria is associated with a very high risk of preterm delivery (and, even more importantly for the potential effect on the child’s health, of early preterm delivery), and of deterioration in kidney function [3,4,5,6,7].

In these patients, delivery is often induced before term on the basis of maternal (deterioration of the kidney function) or foetal problems (alterations in utero-placental Doppler flows), the two being frequently associated; different policies may partially account for the different results observed world-wide. Sometimes, but not invariably, maternal or foetal problems are associated with clinical features that suggest superimposed preeclampsia (worsening of hypertension and proteinuria). The fact that preterm delivery in CKD patients is often associated with normal foetal growth is particularly challenging, as it suggests the presence of complex interactions between the kidney and the placenta that differ from the classical defect in placental implantation which characterizes forms of “placental” preeclampsia, where foetal growth restriction is a common associated finding [25,26]. In line with this observation, the pattern of alteration of the balance between pro-angiogenic and anti-angiogenic biomarkers is probably different in CKD pregnancies or in preeclampsia superimposed on CKD, with respect to preeclampsia in the absence of CKD, thus limiting the prognostic value of these tests in a case such the one described here [27,28].

There is a large body of evidence that suggests that kidney-function impairment during pregnancy and immediately afterwards is more common in advanced CKD stages than in earlier stages. However, the data reported on short-term CKD progression vary from 20 to 80% of cases, too wide a range to allow a precise estimate of the risk in an individual patient [4].

Furthermore, we can postulate that the effect of pregnancy-associated hyperfiltration is higher in cases, such in our patient, in which kidney disease is at least partly related to hemodynamic stress on the remnant nephrons [29]. However, we lack the data to support clinical logic and, while scattered experience suggests that comprehensive clinical management including dietary counselling and moderate protein restriction may be of help in controlling proteinuria, the evidence is too scant to allow quantitative conclusions [30,31,32,33].

It should be noted that pregnancy is often complicated in MELAS patients [34,35,36,37,38,39].

The high incidence of hypertensive disorders of pregnancy, including preeclampsia, reported in MELAS patients, may be an indirect effect of the progressive nephron loss, as these alterations are associated with CKD pregnancies, since their earlier stages, and are described in the case of reduction of kidney tissue, including after kidney donation, where they may reflect a particular response of a low nephron number to the pregnancy challenge [27,28,34,35,36,37,38,39,40,41].

In our patient, the uncertainty in pregnancy outcomes and in the renal prognosis was accompanied by an unpredictable neurologic prognosis of her mitochondrial disease, which had already shown signs of organ involvement (diabetes, kidney disease, deafness, cardiac hypertrophy). If she were to give birth, it would be impossible to predict the disease phenotype the baby might develop [41,42,43,44].

In fact, the baby would have to face two major challenges, one related to the high probability of prematurity and one to inheriting its mother’s disease. Taken together the long-term prognosis was poor. From the renal point of view, prematurity is associated with a reduced nephron number and this finding is the basis of the increased risk of hypertension and CKD found in children born preterm and/or with low to very low birth weight [8,10]. In this regard it is conceivable that the effect of the microvascular disease of MELAS syndrome would be amplified in the context of a low nephron patrimony. Likewise, prematurity is associated with a higher risk of intellectual deficits and, once more, an individual with a suboptimal background may be more prone to develop symptomatic effects of the inherited disease [45].

This said, the association between preterm or early preterm delivery and the development of diseases later in life has not yet been proven to be inevitable, and an increased risk of disease in adulthood is still associated with an absence of evident clinical derangements in most individuals. Furthermore, even though the example given above of Nietzsche, one of the geniuses of the nineteenth century, focuses on a grim, turbulent medical history, his story also exemplifies the fact that extraordinary mental ability lasting several decades is not incompatible with this syndrome, and the late development of the clinical disease leaves hope in future medical developments, and in the possibility of developing effective treatment [16,46,47,48].

## 5. Conclusions

The case described shows how important it is to consider mitochondrial disorders in the differential diagnosis of focal segmental glomerulosclerosis, in patients who display an array of other signs and symptoms, such as short stature, diabetes, cardiac hypertrophy, neurologic involvement, or sensoneural deafness, and in particular in the presence of a family history of CKD, diabetes, and vascular disorders, especially if such diseases are present in the mother. Since kidney involvement is not limited to focal segmental glomerulosclerosis, the diagnosis of MELAS-associated nephropathy should be born in mind, in the presence of interstitial nephropathies or nephroangiosclerosis, especially if of early onset, in the context of any combination of diseases mentioned above [21].

Indeed, the role of mitochondrial diseases in kidney involvement may be higher than previously thought, and next-generation approaches are now allowing sequencing of the entire mtDNA, at a relatively low cost, also employing cells present in the urinary sediment, thus allowing a rapid diagnosis. Higher awareness may lead to a better quantification of the role of the MELAS and related syndromes as a cause of kidney diseases, thus filling a present knowledge gap. Considering the spectrum of renal involvement in different mitochondrial inherited diseases, an accurate clinical phenotyping is still fundamental to address the analysis [49].

Furthermore, this case highlights the difficulties in counselling the prospective mother, in the presence of multiple uncertainties for both mother and child in the short- and long-term prognosis of MELAS syndrome.

Specifically from the renal point of view, we have to acknowledge the limits of our knowledge about possible interactions between prematurity, retarded and impaired nephron development, and MELAS syndrome in determining the future health of the baby, and the difficulty of foreseeing the effect of pregnancy-related hyperfiltration on the progression of kidney disease in the mother.

## Figures and Tables

**Figure 1 jcm-08-00303-f001:**
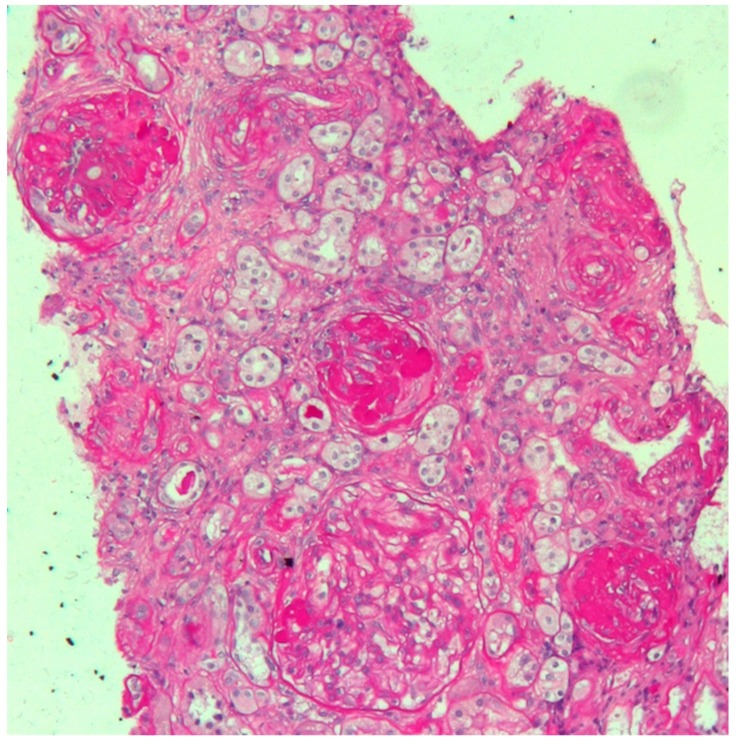
Kidney biopsy. Periodic Acid Schiff (PAS) staining, 20X: light microscopy showing three sclerotic glomeruli and a segment of sclerosis in a large hypertrophic glomerulus. Interstitial oedema with mild inflammatory infiltrate is also evident. Arterioles show marked thickening of tunica media.

**Figure 2 jcm-08-00303-f002:**
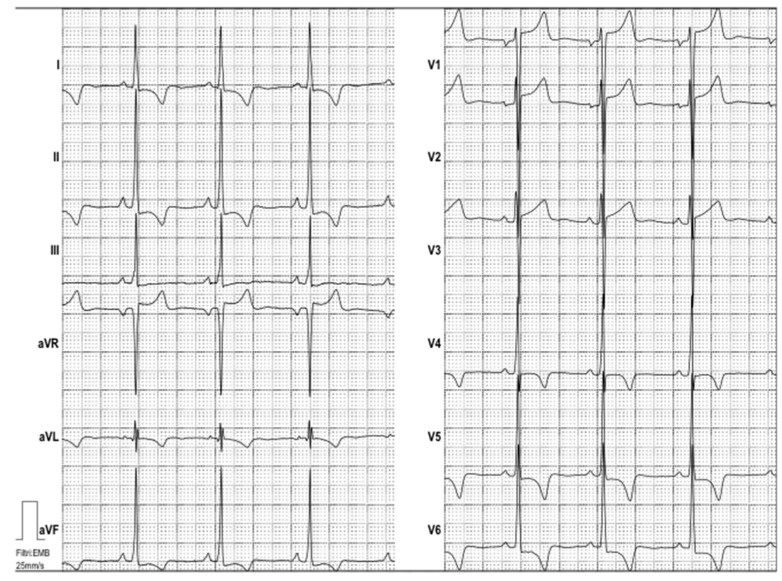
Electrocardiogram. Electrocardiogram showed sinus rhythm and high voltages of QRS with inverted T waves both in the peripheral and precordial leads, suggesting left ventricular hypertrophy.

**Figure 3 jcm-08-00303-f003:**
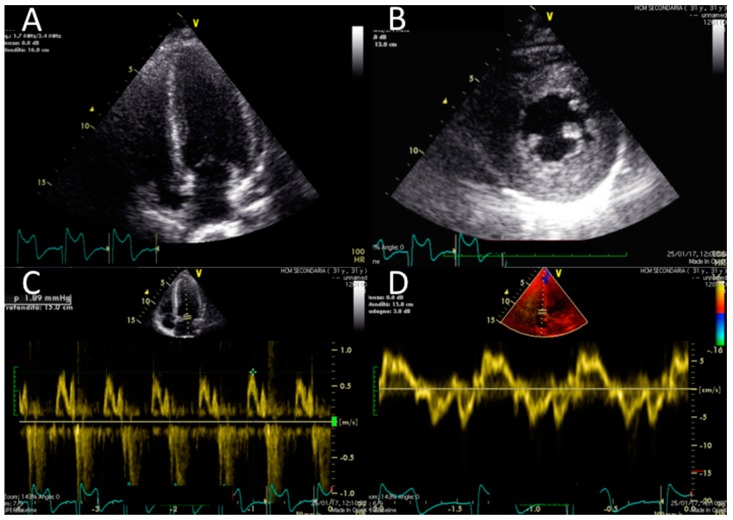
Echocardiographic findings. Echocardiographic findings show a moderate increase in LV thickness (max diameter 17 mm inferior and inferoseptal walls) in a four-chamber view (panel **A**) and on the mid short axis (panel **B**). Slightly reduced longitudinal function (S wave 0.07 cm/s) (**D**) and no significant abnormality of diastolic function (normal E/A pattern and E/E′ 10) were found (panel **C** and **D**) (Appendix A).

**Figure 4 jcm-08-00303-f004:**
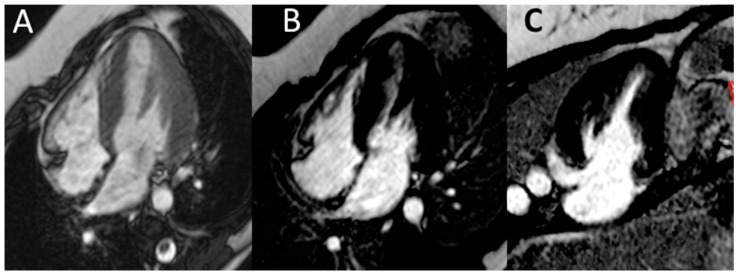
Cardiac magnetic resonance. Cardiac magnetic resonance (CMR) confirmed normal LV volume, systolic function and a moderate increase in LV thickness (panel **A**). No abnormalities in late gadolinium enhancement in the horizontal long-axis view (**B**) or vertical long-axis view (**C**) were found.

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
