# Peer review of "Mitochondrial Disease (MELAS Syndrome) Discovered at the Start of Pregnancy in a Patient with Advanced CKD: A Clinical and Ethical Challenge"

_jcm, 2019, doi:10.3390/jcm8030303_

Reviewer 1 Report

The authors discuss a very interesting case of unexpected pregnancy in patient with MELAS in the context of chronic kidney disease.

I have few comments regarding the manuscript:

Did the authors measure lactic acid in serum?

It would be important to mention the methodology used for the detection of the mitochondrial DNA mutation.

Do the authors follow the patient after the termination of the the pregnancy? If so, how does the termination affect the blood markers of CKD?

The may discuss how different mitochondrial mutations affect kidney function and the different methods to detect mitochondrial mutation. Thus, mitochondrial sequencing is cheaper and easier to performed than it used to be a decade ago.

Author Response

Reviewer 1:

The authors would like to thank the reviewer for the interesting comments and suggestions;

Please find our answers in the following lines:

Comment

The authors discuss a very interesting case of unexpected pregnancy in patient with MELAS in the context of chronic kidney disease.

I have few comments regarding the manuscript:

Did the authors measure lactic acid in serum?

Answer:

Yes, we measured basal lactic acid in serum and after exercise; we found normal basal values and mild lactic acidosis after exercise on cyclergometer. The test was perfomed a few months after pregnancy, and the results were not added in the paper, since the analysis was not performed in pregnancy.

Comment

It would be important to mention the methodology used for the detection of the mitochondrial DNA mutation.

Answer: 

Thanks for the suggestion:  we added this precision: Sanger sequencing of MT-TL1 gene from DNA extracted from blood and urine identified the presence of the classical 3243 A>G mutation.

Comment

Do the authors follow the patient after the termination of the pregnancy? If so, how does the termination affect the blood markers of CKD?

Answer:

Yes, we followed the patient after termination pregnancy. The following information was integrated in the paper. 

After pregnancy termination, therapy with ACEi was started again. Kidney function remained stable up to two years after pregnancy discontinuation (at February 2019: serum creatinine 2.0 mg/dl, proteinuria 2.16 g/24 h).

Comment

The authors may discuss how different mitochondrial mutations affect kidney function and the different methods to detect mitochondrial mutation. Thus, mitochondrial sequencing is cheaper and easier to performed than it used to be a decade ago.

Answer

We agree with the reviewer on this regard, and we added a few sentences on this issue, and add the great review of Finisterer and Scorza as follows: 

The kidney disease associated with MELAS syndrome are protean and may involve at a different degree all structures: vascular (infarction, nephroangiosclerosis), interstitial (Fancony syndrome, complex tubular disorders) and glomerular, where the appearance of focal segmental lesions is frequently associated with progressive kidney failure(17-20). Different kidney neoplastic diseases may also occur (18). 

The kidney biopsy in this case presents the features of glomerular sclerosis and of glomerulomegalia, thus suggesting a form of focal segmental glomerulosclerosis (FSGS), secondary to nephron loss and to hypertrophy of the remnant nephrons (figure 1). 

The renal involvement in mitochondrial disorders is relatively common and may take different forms; heterogeneity is high both in the context of the same mutation (the best known of which is the one presented by our patient, characteristic of the MELAS syndrome), where probably the main determinant is the quantity of mutated DNA in each tissue, and across the different mutations described. 

Two other well-characterised diseases with kidney involvement are Kearns-Sayre syndrome (KSS) characterized by early onset (before 20 years of age) of progressive external ophthalmoplegia and pigmentary retinitis; deafness, cerebellar ataxia and heart block are frequent features, and kidney disease is frequent. In these cases the involvement is typically interstitial, with renal tubular acidosis, Barter syndrome or tubulo-interstitial nephritis, with or without nephrocalcinosis, while focal segmental glomerulosclerosis is rare.  The so-called “Mitochondrial depletion syndromes” are characterised by a reduction in the quantity of mtDNA within a mitochondrion of a cell; these heterogeneous diseases are linked to various mutations, and are likewise associated with interstitial kidney diseases. Kidney failure however, seems rare.

While an extensive review is beyond the scope of this paper, the readers may refer to the brilliant review by Finsterer and Scorza for an extensive discussion on the kidney manifestations of mitochondrial diseases (21). 

We also added a sentence on sequencing in the conclusions, stressing the fact that diagnosis is highly facilitated by the new available technologies. 

Indeed, the role of mitochondrial diseases in kidney involvement may be higher than previously thought, and next-generation approaches are now allowing sequencing of the entire mtDNA, at a relatively low cost, employing also cells present in the urinary sediment, thus allowing a rapid diagnosis. Considering the spectrum of renal involvement in different MID, an accurate clinical phenotyping is still fundamental to address the analysis (47).

With the hope that the revised version has answered to all the questions and comments,

Best regards,

The authors

Reviewer 2 Report

Unique case presentation: MELAS + renal involvement + pregnancy.

Good quality of English and structure.

MELAS is a rare disease - introduction/case would benefit from a more in-depth summary of MELAS-associated kidney disease(s), incidence and outcomes. Perhaps move the information in the Discussion:diagnostic issues to the introduction?

The case - how was pregnancy confirmed, and at what gestation?

Figures 2 - 4 add little to the description given in the text. A video of the echocardiogram might be a useful addition as supplemental data, but static images less helpful.

Row 146: "are" instead of "is"

Row 154-158: One of the author's own papers (Piccoli GB et al. JASN 2010:5;844-855) identified that most preterm delivery in women with CKD is iatrogenic due to "maternal indications" rather than fetal development concerns. Progressive hypertension/proteinuria likely to be maternal/renal in origin rather than placental?

Row 159-162: Role of angiogenic factors of potential interest widely, but not relevant for this case discussion.

Row 200: I agree that mitochondrial disorders are an interesting diagnostic possibility in cases of FSGS, but what is the prevalence of MELAS, and of MELAS-associated nephropathy, compared with the prevalence of FSGS from other causes (idiopathic, obesity, preterm birth, etc). I acknowledge that the recommendation to consider the diagnosis is clarified by mentioning associated clinical features to promote a higher likelihood of diagnosis.

Author Response

Reviewer 2

Comment

MELAS is a rare disease - introduction/case would benefit from a more in-depth summary of MELAS-associated kidney disease(s), incidence and outcomes. 

Perhaps move the information in the Discussion: diagnostic issues to the introduction?

Answer, 

thank you for your comment: we indeed implemented the discussion, as also suggested by your co-reviewer. We added some sentences, and added two references of very good reviews on this issue (ref 21 and ref 47). 

 The kidney disease associated with MELAS syndrome are protean and may involve at a different degree all structures: vascular (infarction, nephroangiosclerosis), interstitial (Fancony syndrome, complex tubular disorders) and glomerular, where the appearance of focal segmental lesions is frequently associated with progressive kidney failure(17-20). Different kidney neoplastic diseases may also occur (18). 

The kidney biopsy in this case presents the features of glomerular sclerosis and of glomerulomegalia, thus suggesting a form of focal segmental glomerulosclerosis (FSGS), secondary to nephron loss and to hypertrophy of the remnant nephrons (figure 1). 

The renal involvement in mitochondrial disorders is relatively common and may take different forms; heterogeneity is high both in the context of the same mutation (the best known of which is the one presented by our patient, characteristic of the MELAS syndrome), where probably the main determinant is the quantity of mutated DNA in each tissue, and across the different mutations described. 

Two other well-characterized diseases with kidney involvement are Kearns-Sayre syndrome (KSS) characterized by early onset (before 20 years of age) of progressive external ophthalmoplegia and pigmentary retinitis; deafness, cerebellar ataxia and heart block are frequent features, and kidney disease is frequent. In these cases the involvement is typically interstitial, with renal tubular acidosis, Barter syndrome or tubulo-interstitial nephritis, with or without nephrocalcinosis, while focal segmental glomerulosclerosis is rare.  The so-called “Mitochondrial depletion syndromes” are characterized by a reduction in the quantity of mtDNA within a mitochondrion of a cell; these heterogeneous diseases are linked to various mutations, and are likewise associated with interstitial kidney diseases. Kidney failure however, seems rare.

While an extensive review is beyond the scope of this paper, the readers may refer to the brilliant review by Finsterer and Scorza for an extensive discussion on the kidney manifestations of mitochondrial diseases (21).

Comment 

The case - how was pregnancy confirmed, and at what gestation?

Answer.

Thanks for the questions, we added the precision:

(self-performed commercial rapid test, at 6 weeks of gestation, later confirmed by echography at 8 weeks of gestation).

Figures 2 - 4 add little to the description given in the text. A video of the echocardiogram might be a useful addition as supplemental data, but static images less helpful.

Answer

We provided a short video of the echocardiogram, which we added to the static images.

Row 146: "are" instead of "is"

Ok thanks, corrected.

Comments

Row 154-158: One of the author's own papers (Piccoli GB et al. JASN 2010:5;844-855) identified that most preterm delivery in women with CKD is iatrogenic due to "maternal indications" rather than fetal development concerns. Progressive hypertension/proteinuria likely to be maternal/renal in origin rather than placental?

Row 159-162: Role of angiogenic factors of potential interest widely, but not relevant for this case discussion. 

Answers:

Indeed, the two points are related in our opinion; we mentioned the angiogenic antiangiogenic factors to highlight how in these cases we cannot rely on them for foreseeing pregnancy outcomes;  we tried to better clarify this issue as follows: 

In these patients, delivery is often induced before term on the basis of maternal (deterioration of the kidney function) or foetal problems (alterations in utero-placental Doppler flows), the two being frequently associated; different policies may partially account for the different results observed world-wide. Sometimes, but not invariably, maternal or foetal problems are associated with clinical features that suggest superimposed preeclampsia (worsening of hypertension and proteinuria). The fact that preterm delivery in CKD patients is often associated with normal foetal growth is particularly challenging, as it suggests the presence of complex interactions between the kidney and the placenta that differ from the classical defect in placental implantation which characterizes forms of “placental” preeclampsia, where foetal growth restriction is a common associated finding (25-26). In line with this observation, the pattern of alteration of the balance between pro-angiogenic and anti-angiogenic biomarkers is probably different in CKD pregnancies or in preeclampsia superimposed on CKD, with respect to preeclampsia in the absence of CKD, thus limiting the prognostic value of these tests in a case such the one described here (27-28).

And in the subsequent lines:

 It should be noted that pregnancy is often complicated in MELAS patients (34-39). 

The high incidence of hypertensive disorders of pregnancy, including preeclampsia, reported in MELAS patients, may be an indirect effect of the progressive nephron loss, as these alterations are associated with CKD pregnancies, since their earlier stages, and are described in the case of reduction of kidney tissue, including after kidney donation, where they may reflect a particular response of a low nephron number to the pregnancy challenge (27-28, 34-41). 

Row 200: I agree that mitochondrial disorders are an interesting diagnostic possibility in cases of FSGS, but what is the prevalence of MELAS, and of MELAS-associated nephropathy, compared with the prevalence of FSGS from other causes (idiopathic, obesity, preterm birth, etc). 

I acknowledge that the recommendation to consider the diagnosis is clarified by mentioning associated clinical features to promote a higher likelihood of diagnosis.

Comment

The reviewer is absolutely right; these are rare diseases, we tried to clarify this better as follows: We also mentioned that these diseases are probably under-diagnosed, no precise figure of  prevalence is therefore available. 

The case described shows how important it is to consider mitochondrial disorders in the differential diagnosis of focal segmental glomerulosclerosis, in patients who display an array of other signs and symptoms, such as short stature, diabetes, cardiac hypertrophy, neurologic involvement, or sensoneural deafness, and in particular in the presence of a family history of CKD, diabetes and vascular disorders, especially if such diseases are present in the mother. Since kidney involvement is not limited to focal segmental glomerulosclerosis, the diagnosis of MELAS-associated nephropathy should be born in mind, in the presence of interstitial nephropathies or nephroangiosclerosis, especially if of early onset, in the context of any combination of diseases mentioned above (21).

Indeed, the role of mitochondrial diseases in kidney involvement may be higher than previously thought, and next-generation approaches are now allowing sequencing of the entire mtDNA, at a relatively low cost, employing also cells present in the urinary sediment, thus allowing a rapid diagnosis. Higher awareness may lead to a better quantification of the role of the MELAS and related syndromes as a cause of kidney diseases, thus filling a present knowledge gap. Considering the spectrum of renal involvement in different MID, an accurate clinical phenotyping is still fundamental to address the analysis (49).

With the hope that the present version may have answered to the questions and comments,

Best regards 

The authors.